# Plasma Metabonomics in Insulin-Resistant Hypogonadic Patients Induced by Testosterone Treatment

**DOI:** 10.3390/ijms23147754

**Published:** 2022-07-14

**Authors:** Lello Zolla, Giuseppe Grande, Domenico Milardi

**Affiliations:** 1University of Tuscia, 01100 Viterbo, Italy; 2Unit of Andrology and Reproductive Medicine, University of Padua, 35122 Padua, Italy; grandegius@gmail.com; 3Division of Endocrinology, Fondazione Policlinico Gemelli, IRCCS, 00168 Rome, Italy; domenico.milardi@policlinicogemelli.it; 4International Scientific Institute, “Paul VI”-Fondazione Policlinico Gemelli, IRCCS, 00168 Rome, Italy

**Keywords:** insulin resistance, testosterone therapy, hypogonadism, ketone bodies, metabolisms, ketosis, lactate

## Abstract

Hypogonadic subjects with insulin resistance (IR) showed different metabonomic profiles compared to normo-insulinemic subjects (IS). Testosterone replacement therapy (TRT) may have a different impact on the metabolisms of those with the presence or absence of insulin resistance. We evaluated the changes in the metabolism of IR hypogonadic patients before and after 60 days of TRT. The metabonomic plasma profiles from 20 IR hypogonadal patients were recorded using ultra-high-performance liquid chromatography (UHPLC) and high-resolution mass spectrometry (HRMS). Plasma metabolites, before and after 60 days of TRT, were compared. In hypogonadic patients, carnosine, which is important for improving performance during exercise, increased. Conversely, proline and lysine—amino acids involved in the synthesis of collagen—reduced. Triglycerides decreased and fatty acids (FFAs) increased in the blood as a consequence of reduced FFA β-oxidation. Glycolysis slightly improved, while the Krebs cycle was not activated. Gluconeogenesis (which is the main energy source for hypogonadal IR before TRT) stopped after treatment. As a consequence, lactate and acetyl CoA increased significantly. Both lactate and acetyl CoA were metabolized into ketone bodies which increased greatly, also due to leucine/isoleucine degradation. Ketone bodies were derived predominantly from acetyl CoA because the reaction of acetyl CoA into ketone bodies is catalyzed by mtHMGCoA synthase. This enzyme is inhibited by insulin, which is absent in IR patients but overexpressed following testosterone administration. Ketosis is an alternative route for energy supply and provides the same metabolic effects as insulin but at the metabolic or primitive control level, which bypasses the complex signaling pathway of insulin. After treatment, the hypogonadic patients showed clinical symptoms related to ketonuria. They presented similarly to those following a ketogenic diet, the so-called ‘keto flu’. This must be taken into account before the administration of TRT to hypogonadic patients.

## 1. Introduction

It is known that serum testosterone (T) levels decrease steadily after 50 years of age. It is predicted that at 70 years of age, the production rate of testosterone decreases to less than 50% of that of a younger male [1,2]. However, it should be noted that occasionally, men may have low testosterone levels at a younger age.

There are two main types of hypogonadism: primary (where the alteration of production is due to testicular defects) and secondary (where the defect could be in the hypothalamus or in the pituitary gland) [3]. Testosterone plays a vital role in regulating sexual behavior, improving cognitive functioning, and changing body composition (increasing the lean body mass and decreasing the fat mass). Particularly, the effects of testosterone during development focus mainly around the penis, testicles, facial and body hair, and muscle growth. Testosterone also regulates physiological libido, the frequency of spontaneous erections, the development of muscle mass, tone of voice, and bone maturation [4]. Many studies conducted on men in the past 30 years have shown that low testosterone levels are related to the incidence of obesity, metabolic syndrome, and cardiovascular disease [5]. An accumulation of fat can lead to insulin resistance and subsequently to diabetes; consequently, IR hypogonadism is often also associated with diabetes [6]. Insulin resistance is defined as the impairment of the insulin-mediated glucose disposal by the body [7,8]. Interestingly, several studies have shown the existence of a bidirectional relationship between low testosterone levels and insulin-resistant states [9,10,11,12,13,14]. In fact, at low testosterone concentrations, a reduced expression of genes such as insulin receptor beta subunit (IR-β), insulin receptor substrate 1 (IRS-1), AKT serine/threonine kinase-2 (AKT2), and solute carrier family 2 member 4 (SLC2A4) or glucose-transporter-type 4 (GLUT4) was observed, all of which mediate the signaling of insulin, responsible for glucose transport [8,15]. Thus, young patients with hypogonadism may initially present normo-insulinemic (IS) states, but over time, their blood insulin concentration may increase, leading to insulin resistance (IR) [16] and type 2 diabetes [17]. The presentation of patients with hypogonadism was compared based on whether they were IS or IR. Both groups showed different clinical complications and different metabolic changes [18,19]. For this reason, in our previous investigations [18,19], hypogonadic patients were classified into insulin-resistant (IR) and insulin-sensitive (IS) categories using their HOMAi (homoeostatic model index for insulin resistance). This highlighted the most impacted pathways in order to better understand the two different subgroups.

Testosterone was introduced into clinical settings as substitution therapy, and its efficacy and safety in the treatment of male hypogonadism was evident [20]. The administration of testosterone gel formulation has become a popular testosterone replacement therapy in patients with hypogonadism since their advent in the year 2000. The use of TRT is still controversial—there are many benefits associated with using TRT, including better bone mineral density, better sexual function, and increased strength. Conversely, recent studies have reported that the use of TRT may carry an associated cardiovascular risk, especially in older men and younger men with heart disease.

Testosterone replacement should only be given to men with a diagnosis of hypogonadism based on persistently low serum testosterone concentrations and symptoms related to low testosterone levels. The aim of testosterone replacement therapy is to increase the serum testosterone level in the medium–normal range and the resolution of or reduction in hypogonadism symptoms [21]. Since testosterone is a hormone influencing metabolomic pathways, it is of interest to know which metabolic pathways were or were not restored upon TRT. To this regard, it is important to distinguish IR patients from IS patients so as to follow the differences in improvement following testosterone therapy.

The aim of the study was to evaluate, in a group of hypogonadal subjects with insulin resistance, how and which pathways are modified after TRT. For this purpose, hypogonadic insulin-resistant patients with HOMAi > 2.5, testosterone levels < 8 nmol/L, high insulin (>18 mUI/L), and high BMI (body mass index) (30.27 ± 2.69) were recruited. Their plasma was analyzed, and then they were treated with gel testosterone for 60 days. It is of note that this group of patients achieved normal testosterone levels after just 40 days of treatment, but the collection of sample data reported below refers to the metabolites measured in the plasma after 60 days.

## 2. Results

### 2.1. Clinical and Hormonal Parameters

The clinical and hormonal results are reported in Table 1. Total testosterone (T) was significantly reduced in the hypogonadic patients compared to the control group (*p*-value < 0.05). After TRT, a significant improvement in T levels was observed. Although the BMI of the hypogonadal subjects was higher than the controls, the statistical analysis did not show significant differences in the three groups.

### 2.2. Metabonomic Analysis

In this study, we compared the metabolic changes in 20 IR hypogonadic patients before and after 60 days of testosterone treatment. We compared this to 20 control subjects.

It is of note that the IR hypogonadic patients showed restored testosterone in the medium–normal range levels after just 40 days of treatment (18.10 ± 5.48 nmol/L), but blood collection was performed after 60 days, to be sure of the metabolic pathway’s effect on restoration and equilibration.

The baseline characteristics of the study subjects and endocrine variables between healthy males and hypogonadal men, before and after testosterone treatment for 60 days, are shown in Table 1. Analysis was performed using biochemical or immunological assays. After TRT, very small differences were observed between HOMAi and BMI (see Table 1). Insulin went down from 17.33 ± 4.88 (mUI/L) to 15.07 ± 3.39 (mUI/L) but did not reach the control value of 7.77 ± 3.13 (mUI/L). Meanwhile, there was a significant increase in testosterone levels, rising from 8.07 ± 4.13 to 18.10 ± 5.48 nmol/L (closer to the control of 20.02 ± 7.47 nmol/L). Triglycerides decreased significantly, while cholesterol, HDL, and VHL did not increase.

Still using the plasma of the IR hypogonadic patients, we conducted an exploratory investigation using non-targeted high-resolution mass spectrometry (HPLC-MS) before and after treatment with testosterone. Using this, we compared the metabolic differences related to testosterone therapy. Thus, the untargeted metabolomic profiling of plasma from the same subjects was performed using online tools (MetaboAnalyst 3.0 software). To identify which metabolic pathways were most impacted in IR hypogonadic males after treatment, we performed an analysis of the overview of pathways according to *p*-values from their enrichment and impact values. Figure 1A shows MESA (Metabolite Set Enrichment Analysis), which is a platform used to identify biologically significant changes in the concentration of metabolites for quantitative metabolomic studies. The MSEA investigated if a group of functionally related metabolites was significantly enriched. The *p*-value indicated the strength of the association between the profiled metabolite and the class label. The results showed that the metabolites most identified were involved in citric acid metabolism, in the Warburg effect, in the transfer of acetyl groups into mitochondria, valine, leucine/isoleucine degradation, and the oxidation of branched chain fatty acids. At the same time, in order to identify the most relevant pathways altered, the data were analyzed using Metabolic Pathways Analysis (MetPA) (Figure 1B), which combines results from the pathway enrichment analysis and the pathway topology analysis. The “metabolic view” is the predicted ratio between the enrichment analysis (*y*-axis) and the topology analysis (*x*-axis), with the most significant *p*-values indicated in red and the least significant values in yellow and white. MetPA analysis predicted that the most significant pathways changed upon TRT were pyruvate metabolism, citrate cycle (TCA), valine, leucine and isoleucine metabolism, purine metabolism, histidine metabolism, arginine and proline metabolism, glycolysis, and gluconeogenesis.

In the collection of single metabolites into the proper metabolic pathways, most predictions from MetPA were confirmed.

### 2.3. Glycolysis

Glycolysis, which was strongly reduced in IR hypogonadic patients [17], was slightly increased after TRT (Figure 2A). Consequently, in the liver of IR hypogonadic patients, gluconeogenesis, the only energy source of IR [19], stopped. As a result, the glycerol shuttle was reactivated to reoxidize the NADH produced by glycolysis. Consequently, 3P-glycerol was not converted into glycerol and incorporated into triglycerides (Figure 2B), which decreased (Table 1), and the levels of dihydroxyacetone post-therapy were lower (Figure 2B). Interestingly, lactate increased significantly after TRT, one of the final products of glycolysis (Figure 1A).

### 2.4. Acetyl-CoA

Acetyl-CoA was strongly up-regulated. A small concentration of it bio-transformed into mevalonic acid (Figure 3), producing a low concentration of cholesterol (Table 1), while it was not used to produce acetylcarnitine (Figure 3A), which is essential to import fatty acids into mitochondria (restoration of β-oxidation of fatty acid was not restored upon testosterone therapy).

Of note, most of the acetyl CoA was used to produce ketone bodies, such as acetoacetate and 3-hydroxybutyrate. A major part of acetyl CoA comes from the degradation of leucine/isoleucine in favor of ketone body formation (Figure 3B). A small percentage of acetyl CoA entered into the Krebs cycle (TCA), which remained inactive (Figure 3C) (as revealed by the low amount of citrate produced (Figure 4)). Moreover, oxaloacetate (OAA), the main precursor of gluconeogenesis, was not produced (Figure 4) from amino acids, indicating that gluconeogenesis, the main energy source in IR hypogonadism [17] was no longer active upon TRT. As a result, although the glycerol shuttle was re-activated upon TRT, the total NAD and NADH was significantly low (Figure 5A), as well as ATP levels (Figure 5B).

### 2.5. Amino Acids

Regarding amino acids, TRT induced a slight decrease in cysteine and tyrosine, while asparagine and tryptophan decreased significantly (see Appendix A). In IR hypogonadism, these amino acids were increased by about 50–70% and used to produce OAA, a crucial Krebs’ intermediate for gluconeogenesis. After TRT, they decreased, supporting a halted gluconeogenesis. To support this, leucine and isoleucine decreased while valine accumulated, supporting the concept that gluconeogenesis was now blocked, since in IR, it was fueled by the degradation of valine to propionyl CoA and finally into OOA, a key precursor of gluconeogenesis.

Finally, proline and lysine significantly reduced after treatment (Figure 5C), indicating their use for collagen fibers’ formation. Finally, carnosine, as well as histidine and uracil, from which the latter is derived, was restored (Figure 6), as already provided using MetaboAnalyst (Figure 1A).

### 2.6. Clinical Observations

From a clinical point of view, after 60 days of testosterone therapy, patients complained of new symptoms, supporting the concept that a complete restoration of wellness was not reached.

## 3. Discussion

Many studies have been conducted to evaluate the role of testosterone replacement treatment (TRT) in hypogonadal men, but only in terms of putative improvements in clinical symptoms, such as swelling of the sebaceous glands, increased libido, increased frequency of erections, increased muscle mass, increased voice, increased height, bone maturation, etc. [22]. On the contrary, little is known about how metabolomic changes in these patients can change before and after testosterone therapy, especially given that it is these metabolisms which are at the base of the aforementioned clinical symptoms. In particular, it is of general interest to determine whether all the metabolic pathways are completely restored by TRT, and if so, when this occurs and if it occurs before any clinical symptoms improve. In this regard, there are conflicting ideas about the role of TRT in hypogonadal males, ever since the efficacy of androgen replacement therapy was disputed in some hypogonadism patients [23]. Therefore, it remains crucial to distinguish between patients with IS hypogonadism and IR hypogonadism. As revealed here, upon the administration of TRT in IR hypogonadic patients for three months, testosterone levels can be restored, but insulin is reduced (decreasing from 17 to 15 µU/mL), in agreement with Kapoor [16]. Since synergic and/or antagonist action between testosterone and insulin exists [24,25], it is not surprising that this partial insulin reduction can limit the total restoration of all metabolisms, and thus, it is of interest to know what is changed upon TRT. By comparing the metabolites in the plasma of IR hypogonadic patients before and after 60 days of TRT treatment, we evaluated which beneficial metabolic effects were as a result of TRT therapy.

Our HRMS metabonomic analysis revealed that approximately 20 canonical biochemical pathways were affected, among which, 12 pathways were implicated to a significant extent (Figure 1).

It should be noted that complete restoration was achieved for only a few metabolites, but not all. Proline and lysine, two amino acids involved in the synthesis of collagen, significantly increased in IR hypogonadic patients [19]. As a direct consequence, after TRT, a decreased synthesis of collagen occurred, supporting the findings of Wang et al. [26]. In IR hypogonadic patients, carnosine (important for improving performance during exercise [27]) was low [19], as well as histidine. Moreover, this significantly increased upon TRT.

Regarding lipid metabolism, lower glycerol levels reduced the triglycerides’ formation, increasing the number of fatty acids (FFAs) which were released in the blood upon TRT (see Table 1). Of note, the FFA of IR patients also increased because they were not bio-transformed into ketone bodies by b-oxidation (Figure 3) because of the reduced acetylcarnitine, which is essential for their translocation into the mitochondrion.

Regarding glucose metabolism, in IR hypogonadic patients, glycolysis was the most down-regulated pathway [19], but, after TRT, a slight improvement in it was recorded. This can be attributed in part to three main causes. The first is the effect of testosterone on the expression and translocation of both the glucose-transporter Glut4 [28] and insulin receptors. The second is its positive action on key enzymes involved in glycolysis [17]. Finally, the improvement was a result of metabolomic reprogramming (see later).

In fact, glucose is only minimally used for energy supply and cannot be sufficient for the complete energy requirements. Interestingly, lactate (one of the end products of glycolysis) was significantly higher after TRT (Figure 2A). Recently, an increase in lactate was associated with type 2 diabetes and insulin resistance [29]. In agreement, in rodent models, it was found that T deficiency—induced by progressive stages of diabetes mellitus—impairs glucose metabolism, favoring metabolic reprogramming toward glycogen synthesis [30]. Gluconeogenesis is strongly active in IR hypogonadic patients [19] but is stopped upon TRT, as revealed here. However, upon TRT, lactate increases, independently from insulin activity, given that it was also observed following TRT in IS hypogonadic patients [31]. This is probably related to the fact that lactate and testosterone cause reciprocal effects in Leydig cells [32,33], where lactate stimulates testosterone production and testosterone stimulates lactate production. Thus, a lactate increase represents the first response to testosterone supplementation. This was also demonstrated by Enoki et al. [34], who showed that testosterone induces an increase in lactate monocarboxylate transporters in rat skeletal muscle. As a consequence of a lactate increase, lactate dehydrogenase exhibits feedback inhibition; therefore, the rate of conversion of pyruvate to lactate is decreased, and for the most part, pyruvate is converted into acetyl CoA by PDH enzymes, which are not down-regulated by insulin [35]. This explains why acetyl CoA increased significantly (Figure 3). A further contribution to the increase in acetyl CoA comes from the increased degradation of leucine/isoleucine (Figure 3B), two well-known ketogenic amino acids [36,37]. Since leucine/isoleucine and valine account for nearly 35% of the essential amino acids in muscle proteins, after TRT, a higher protein catabolism of skeletal proteins was underway, causing a decrease in muscle mass. This complements a recent metabolome profiling study that compared obese and lean human subjects. It revealed a BCAA-related metabolite signature that is correlated with insulin resistance [38,39], which still remains after TRT, explaining the muscle weakness reported by all patients (see Table 2).

Thus, TRT induced a significantly higher production of acetyl CoA both in IR and IS [31] hypogonadic patients. It is of note that the increased acetyl CoA did not enter the Krebs cycle (Figure 3C) in both groups of patients, but instead was used in other pathway metabolisms. Furthermore, acetyl CoA remained low as it was not converted into mevalonic acid. A minor increase in cholesterol production was recorded (Table 1), in agreement with Zitzmann [40]. Acetyl CoA was also not used to produce acetylcarnitine, which thus decreased, blocking the b-oxidation of FFAs (Figure 3A), in agreement with Fukami et al. [41].

Acetyl CoA was preferentially biotransformed into the ketone bodies acetoacetate and 3-hydroxybutyrate (Figure 3B). This reaction was catalyzed by 3-hydroxy-3methylglutaryl-Coa synthase (mtHMGCoA synthase), an enzyme which is inhibited by insulin and overexpressed by testosterone [42]. The induction of ketone body (KB) production upon TRT agrees with the remote evidence that the inhibition of testosterone production in Leydig cells by ethanol [43] also induced a decrease in 3-hydroxybutyrate.

Figure 7 summarizes metabolic pathways induced by testosterone therapy (not dashed lines), while dotted lines show the inactivated pathways. Thus, the production of KB is exclusive to IR hypogonadism, in which insulin is inactive and cannot inhibit ketogenesis. This is for the most part due to the prevention of the breakdown of triglycerides into FFAs and glycerol [44,45], as well as Cori cycle activation, as revealed in TRT of IS-hypogonadism [31].

It is of note that ketone bodies were produced only in IR [19] hypogonadism, and more so upon TRT, while their production was not observed in IS hypogonadism both before and after TRT [31], suggesting that insulin plays a role. On the other hand, the permanence of insulin resistance upon TRT compromises the correct use of glucose, and it is reasonable to suggest that ketone bodies can offer an alternative route, which acts directly on the same metabolic pathways, without requiring the action of the complex signaling insulin pathways. KB increases mitochondrial acetyl CoA concentration by bypassing the PDH complex and instead providing acetyl CoA from acetoacetyl CoA. Moreover, the ketone bodies acetoacetate and d-b-hydroxybutyrate are in near equilibrium with the free mitochondrial [NAD+]/[NADH] ratio in a reaction catalyzed by d-b-hydroxybutyrate dehydrogenase [46]. Thus, both insulin and ketones have the same effects on both the metabolites of the first one-third of the citric acid cycle and on mitochondrial redox states. These both increase the hydraulic efficiency of the well-perfused working heart [47]. The hydraulic efficiency of the heart is 28% greater via the metabolism of ketone bodies compared with a heart that metabolizes glucose alone.

The fundamental reason for this is because there is an inherently higher heat of combustion in d-b-hydroxybutyrate compared to pyruvate, the mitochondrial substrate which is the end product of glycolysis.

However, recently, lactate and b-hydroxybutyrate were indicated as intermediates of energy-metabolism-regulating cellular functions by controlling metabolic, immune, and other body functions [48]. Ketone bodies are utilized as an energy source by partially replacing glucose in a diabetic human’s heart [49].

From a clinical point of view, testosterone therapy does not completely restore the metabolisms and wellness of a patient, since many symptoms are still present after treatment. Table 2 lists the symptoms before and after T treatment. Of particular interest, the new symptoms are similar to those recorded in patients fed a ketogenic diet, known as the so-called ‘keto flu’ [50]. Patients most notably present with psychiatric problems [51]. Thus, when implementing testosterone therapy, one must take into account whether the patient is or is not insulin-resistant. In summary, in the case of IR hypogonadic patients, the management used for keto flu can be applied and taken into consideration for those additional treatments.

## 4. Materials and Methods

### 4.1. Ethics Statement

This study was approved by the local ethics committee (Università Cattolica del Sacro Cuore, Rome, Italy), protocol P/740/CE/2012.

Written, informed consent was provided by all the subjects, and all experiments were conducted according to the Declaration of Helsinki—Ethical principles for medical research involving human subjects.

### 4.2. Patients’ Samples: Study Design and Participants

Human blood plasma samples were collected in accordance with ethical guidelines and approved standard clinical protocols after overnight fasting. EDTA plasma was prepared via 10 min of centrifugation at 4 °C and 3000 g. We evaluated 20 hypogonadal male patients and 20 age- and BMI-matched controls (Table 1). All subjects enrolled were informed about the study protocol and gave their written consent. The diagnosis of hypogonadism was based on the presence of clinical symptoms related to this condition (e.g., delayed sexual development, reduced libido, or erectile dysfunction) and on the results of standard hormonal exams (total testosterone < 8 nmol/L). The patients affected by hypogonadism were only included if they had HOMAi > 2.5. The participants in the control group were healthy males who were referred to the Outpatient Clinic of Endocrinology and Metabolism for check-ups.

### 4.3. Study Treatment

The study was carried out at Fondazione Policlinico A. Gemelli IRCCS, Rome, Italy. Twenty healthy men and twenty men with hypogonadism diagnosed using both clinical symptoms of hypogonadism, including erectile dysfunction, decreased libido, and/or decreased energy, as well as evidence of low serum T (≤8 nmol/L) were enrolled in our study. The hypogonadal patients were treated with testosterone preparation Gel 2%. The gel was formulated to have a similar application and appearance. Serum testosterone concentrations were measured at 60 days. All patients gave their informed consent before participating in the study.

### 4.4. Plasma Collection and Metabolite Extraction

Metabolites were extracted by adding 200 µl of each plasma sample to 600 µL of cold (−20 °C) chloroform: methanol: water (1:3:1 ratio). Samples were vortexed for 1 min and left on ice for 2 h for complete protein precipitation. The solutions were then centrifuged for 15 min at 15,000× *g*.

### 4.5. UHPLC-HRMS

Twenty microliters of extracted plasma were injected into an ultra-high-performance liquid chromatography (UHPLC) system (Ultimate 3000, Thermo) and run in positive mode. Samples were loaded onto a Reprosil C18 column (2.0 mm × 150 mm, 2.5 μm—Dr Maisch, Germany) for metabolite separation. Chromatographic separations were achieved at a column temperature of 30 °C and flow rate of 0.2 mL/min. For positive ion mode (+) MS analyses, a 0–100% linear gradient of solvent A (ddH_2_O, 0.1% formic acid) to B (acetonitrile, 0.1% formic acid) was employed over 20 min, returning to 100% A in 2 min and a 6 min post-time solvent A hold. Acetonitrile, formic acid, and HPLC-grade water and standards (≥98% chemical purity) were purchased from Sigma Aldrich. The UHPLC system was coupled online with a mass spectrometer Q Exactive (Thermo) scanning in full MS mode (2 μscans) at 70,000 resolutions in the 67 to 1000 m/z range, with a target of 1106 ions, a maximum ion injection time (IT) of 35 ms, 3.8 kV spray voltage, 40 sheath gas, and 25 auxiliary gas, operated positive ion mode. Source ionization parameters were spray voltage, 3.8 kV; capillary temperature, 300 °C; and S-Lens level, 45. Calibration was performed before each analysis against positive- or negative-ion-mode calibration mixes (Piercenet, Thermo Fisher, Rockford, IL, USA) to ensure sub-ppm error of the intact mass. Metabolite assignments were performed using computer software (Maven, 18 Princeton, NJ, USA) upon the conversion of raw files into the mzXML format through MassMatrix (Cleveland, OH, USA).

### 4.6. Metabonomic Data Processing and Statistical Analysis

Raw files of replicates were exported, converted into the mzXML format through MassMatrix (Cleveland, OH, USA), and then processed using MAVEN software (http://maven.princeton.edu/) (accessed on 1 April 2013). Mass spectrometry chromatograms were elaborated for peak alignment, the matching and comparison of parent and fragment ions, and tentative metabolite identification (within a 2 ppm mass-deviation range between observed and expected results against the imported KEGG database). To further explore the metabolic differences between the two groups of subjects, multivariate statistical analyses were employed on an MS data set consisting of 20 hypogonadal men pre- and post-treatment. Multivariate statistical analyses were performed on the entire metabonomic data set using the MetaboAnalyst 3.0 software, which also enabled an overview of the data variance structure in an unsupervised manner. Before the analysis, raw data were normalized by median and auto-scaling to increase the importance of low-abundance ions without the significant amplification of noise. The web- based tools MSEA (Metabolite Set EnrichmentAnalysis) and MetPA (Metabolomic Pathway Analysis), which are incorporated into the MetaboAnalyst platform, were used to perform metabolite enrichment and pathway analyses, respectively. Data for metabolites detected in all samples were submitted into MSEA and MetPA with annotation based on common chemical names. Accepted metabolites were verified manually using HMDB, KEGG, and PubChem DBs. A Homo sapiens pathway library was used for pathway analysis. Global test was the selected pathway enrichment analysis method, whereas the node importance measure for topological analysis was the relative betweenness centrality. For MSEA metabolites, data were mapped according to HMDB, and the “metabolite pathway associated metabolites set” library (currently 88 entries) was chosen for the enrichment analysis, which was performed using the package global test. The results were graphed with Graphpad Prism 5.01 (Graphpad SoftwareInc) (CA, USA). Statistical analyses were performed with the same software. Data are presented as mean ± SEM of fold-change relative to the metabolite levels in controls. Differences were considered statistically significant at *p* < 0.05 and further stratified to *p* < 0.01 and *p* < 0.001, respectively.

## 5. Conclusions

In conclusion, our investigation confirms that in insulin-resistant hypogonadism, testosterone therapy does not re-establish complete metabolism restoration. At present, we cannot exclude the fact that those 60 days of TRT are not sufficient; further research would need to be conducted to investigate this. Longer treatment times are most likely necessary to re-establish complete metabolism restoration, although insulin resistance cannot improve with TRT, which is at the heart of the complications observed. Our investigations revealed that upon the application of common testosterone therapy, it is beneficial to add supplementary biochemical markers which were not restored by TRT to obtain a better wellness recovery, such as carnosine, acetylcarnitine, leucine, and isoleucine. Regarding the new symptoms observed after TRT, they could be managed by the assumption of remedies proposed for ‘keto flu’ [49], such as an increase in sodium intake supplements with electrolytes, drinking broth (including bone broth and stock cubes), increasing magnesium, increasing potassium, increasing dietary fats (including avocado, MCT, olives, butter, nuts, and fat bombs), and increasing water intake.

## Figures and Tables

**Figure 1 ijms-23-07754-f001:**
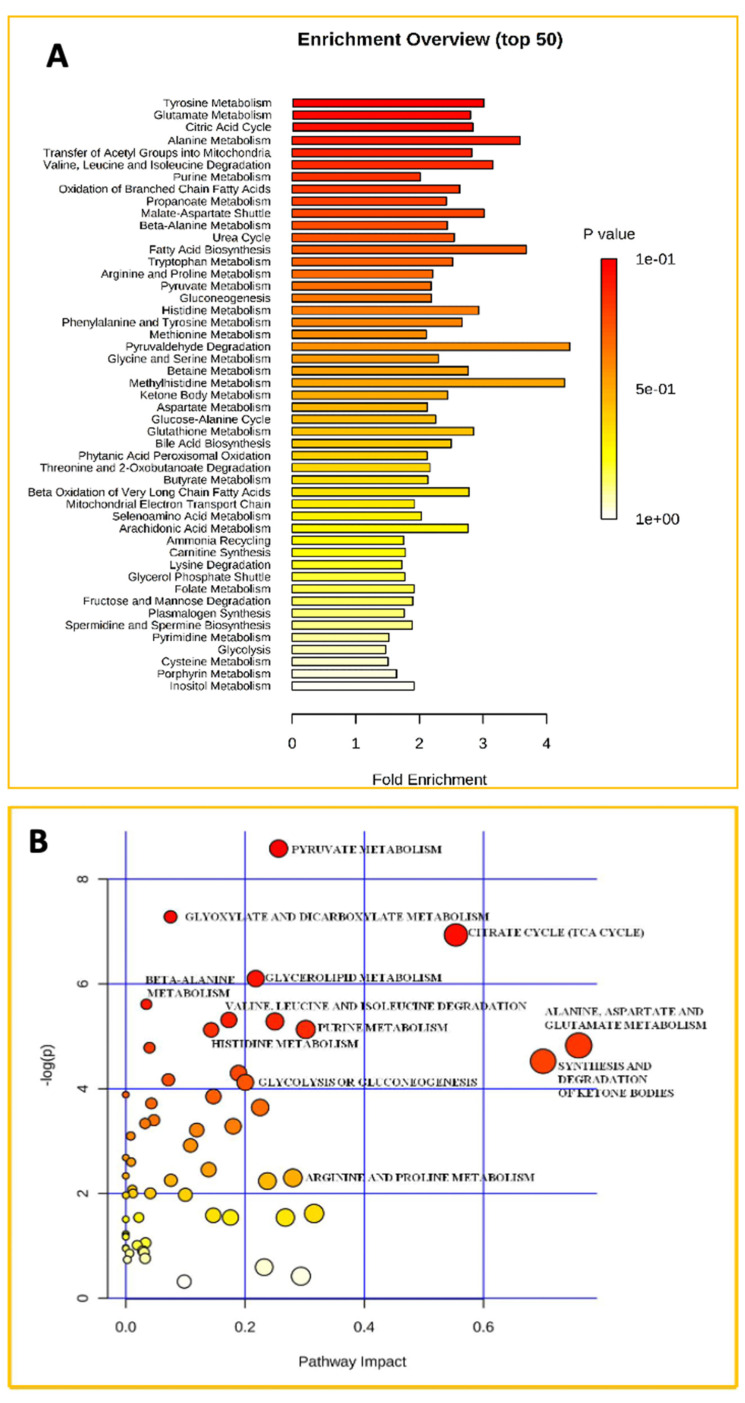
Panel (**A**) Metabolic set enrichment analysis showing the most altered metabolisms as revealed in the plasma of hypogonadal men before and after testosterone replacement treatment (TRT). Color intensity (white to red) reflects increasing statistical significance, while the circle diameter covaries with pathway impact. The graph was obtained by plotting −log of *p*-values from pathway enrichment analysis on the *y*-axis and the pathway impact values derived from pathway topology analysis on the *x*-axis. Panel (**B**) Metabolic Pathway Analysis (MetPA). All the matched pathways are displayed as circles. The color and size of each circle is based on the *p*-value and pathway impact value, respectively. The graph was obtained by plotting on the *y*-axis the −log of *p* values from the pathway enrichment analysis and on the *x*-axis the pathway impact values derived from the pathway topology analysis.

**Figure 2 ijms-23-07754-f002:**
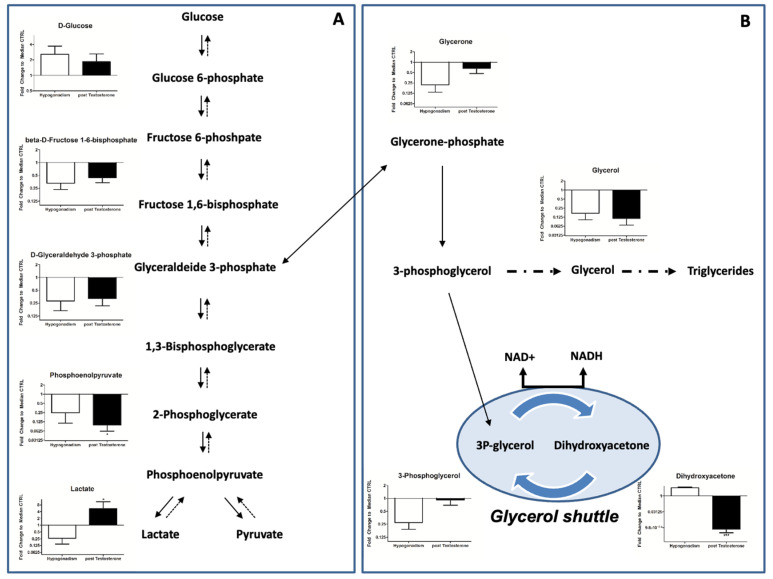
Panel (**A**) Intermediates of glycolysis. The total amount of glycolytic metabolites in the plasma appeared to be slightly restored after testosterone replacement treatment. Panel (**B**) Intermediates involved in glycerol shuttle and triglyceride synthesis. 3P-glycerol was not converted into glycerol to triglyceride biosynthesis, but it was involved in glycerol shuttle, activating it. All data are shown as mean ± SEM of fold-change relative to the metabolite levels in controls. * *p* < 0.05, *** *p* < 0.001.

**Figure 3 ijms-23-07754-f003:**
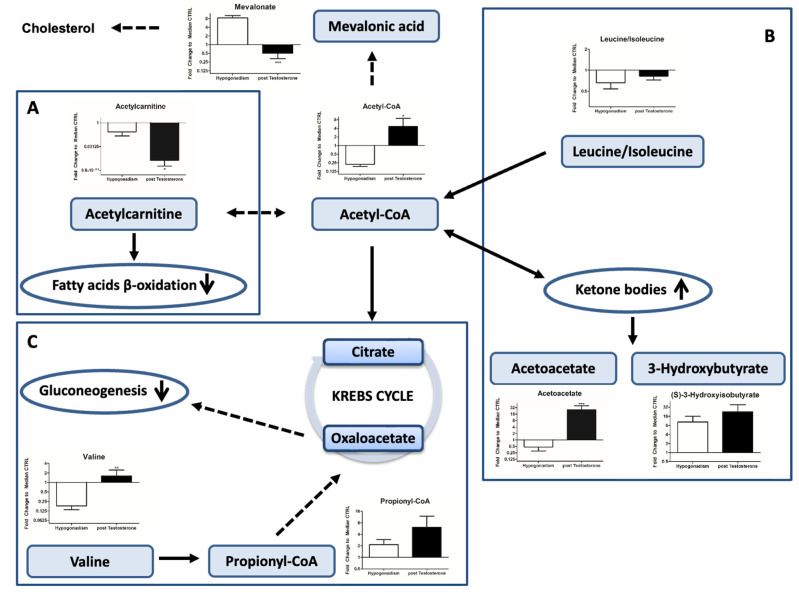
Metabolisms involved in acetyl CoA consumption or production as the main metabolite in glycolytically altered metabolism. Acetyl CoA levels were increased after TRT; instead, low level of acetylcarnitine reported decreased β-oxidation Panel (**A**). Acetyl CoA was mainly converted into ketone bodies Panel (**B**). Finally, high levels of valine and propionyl CoA with low level of oxaloacetate justified decreased gluconeogenesis Panel (**C**). All data are shown as mean ± SEM of fold-change relative to the metabolite levels in controls. * *p* < 0.05, ** *p* < 0.01, *** *p* < 0.001.

**Figure 4 ijms-23-07754-f004:**
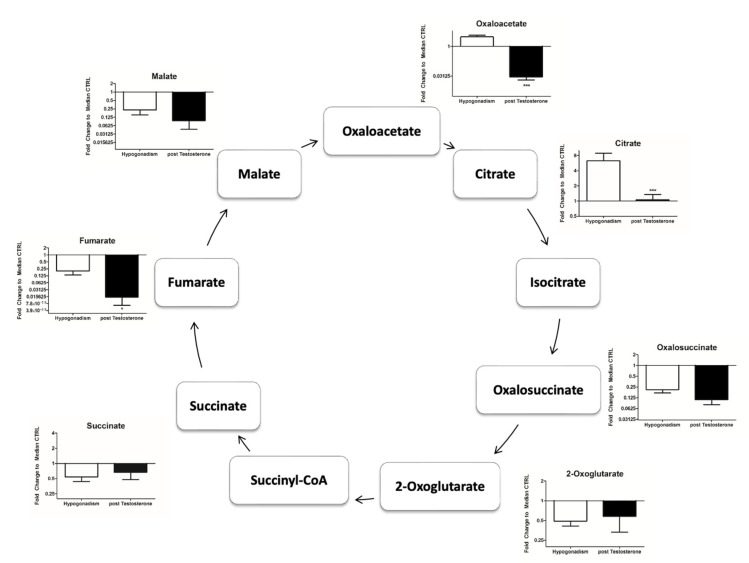
Intermediates of TCA measured in the plasma of hypogonadal patients before and after TRT, revealing that this metabolic pathway was still reduced after therapy. All data are shown as mean ± SEM of fold-change relative to the metabolite levels in controls. * *p* < 0.05, *** *p* < 0.001.

**Figure 5 ijms-23-07754-f005:**
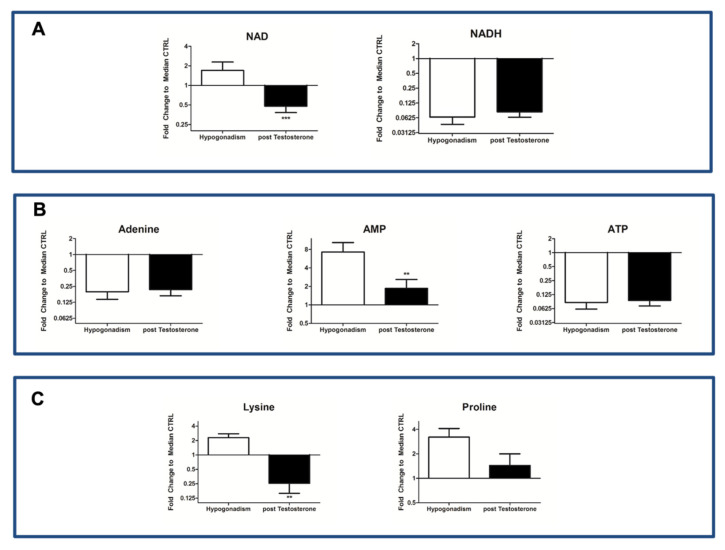
NAD and NADH changes after TRT in hypogonadism Panel (**A**). Down-regulation of TCA cannot restore NAD reduction. Adenine, AMP, and ATP levels were not restored after therapy Panel (**B**). Proline and lysine, amino acids involved in collagen biosynthesis, pre and after TRT Panel (**C**). All data are shown as mean ±SEM of fold-change relative to the metabolite levels in controls. ** *p* < 0.01, *** *p* < 0.001.

**Figure 6 ijms-23-07754-f006:**
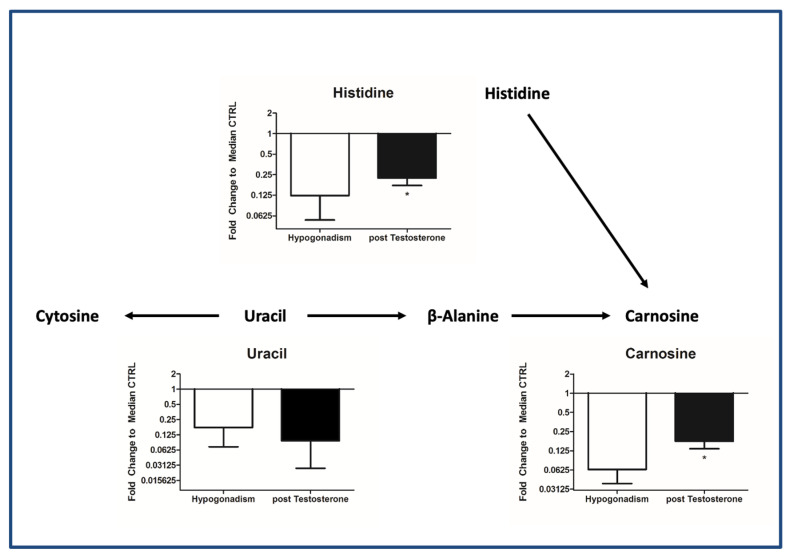
Metabolism of the production of carnosine from β-alanine. All data are shown as mean ± SEM of fold-change relative to the metabolite levels in controls. * *p* < 0.05.

**Figure 7 ijms-23-07754-f007:**
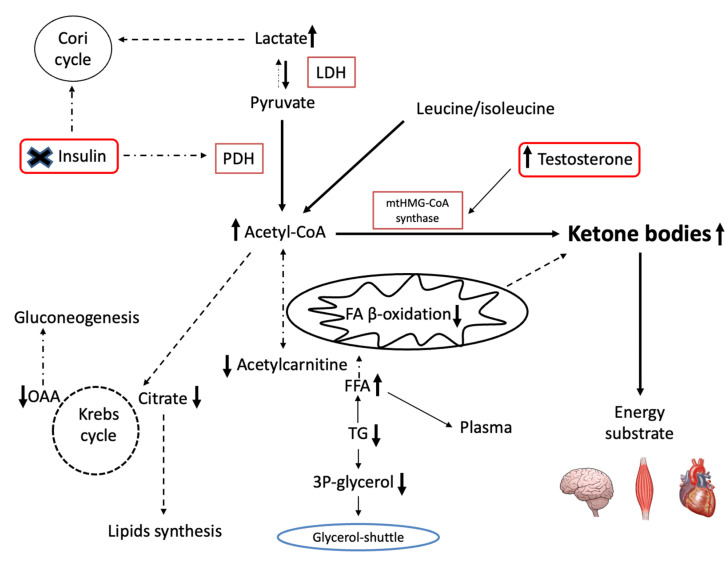
Summary of testosterone therapy effect on most metabolism pathways. In the dashed line the reduced metabolisms are shown, while in the not dashed line, the positively influenced metabolisms are shown. Arrows indicate increases or decreases.

**Table 1 ijms-23-07754-t001:** Characteristics of study participants. BMI: body mass index, TG: triglycerides, LDL: low-density lipoprotein, HDL: high-density lipoprotein. Data are presented as the mean ± SD. Statistical differences were determined using Tukey’s multiple comparisons. ** *p* < 0.01, *** *p* < 0.001.

	Control (A)	IR Hypogonadic (B)	IR Hypogonadic after TRT (C)	*p*-Value	Tukey HSD *p*-Value
**Subjects**	n-20	n-20	n-20	-	-
**Age**	42.60 ± 14.41	49.13 ± 13.01	49.18 ± 16.01	-	-
**BMI (Kg/m^2^)**	23.94 ± 2.54	30.48 ± 3.04	30.50 ± 3.05	0.93	-
**Testosterone (nmol/L)**	20.87 ± 7.37	7.35 ± 3.05	18.10 ± 5.10	0.0001 *** *p*	(A vs. B) 0.001 ** *p* (B vs. C) 0.001 ** *p*
**Glucose (mg/100 mL)**	94.72 ± 4.38	106.13 ± 4.51	99.90 ± 6.77	0.3	-
**Insulin (mUI/L)**	7.07 ± 2.10	17.82 ± 4.88	15.09 ± 2.98	0.69	-
**HOMAi**	1.79 ± 0.86	1.97 ± 0.67	1.47 ± 0.70	0.92	-
**Tg (mmol/L)**	87.8 ± 45.39	227.90 ± 62.73	215.54 ± 63.18	0.5	-
**Cholesterol (mmol/L)**	203.4 ± 29.18	235.81 ± 42.6	210.18 ± 53.18	0.6	-
**HDL Cholesterol (mmol/L)**	55.90 ± 1.98	42.63 ± 15.41	47.45 ± 15.47	0.54	-
**LDL Cholesterol (mmol/L)**	129.45 ± 33.07	142.36 ± 38.74	127.36 ± 44.31	0.85	-

**Table 2 ijms-23-07754-t002:** Glossary of symptoms recorded before (A) and after (B) testosterone therapy.

Glossary of Pre-TRT Symptoms	Glossary of Post-TRT Symptoms	Glossary of Keto flu Symptoms
Sexual dysfunction	Depression Difficulty concentrating	Headache Anxiety Irritability Insomnia Fatigue Decreased energy Depression Difficulty concentrating Depressive symptoms Depressed mood Brain fog Increased body fat Memory impairment Muscolar soreness Muscolarweakness Decreased motivation Nausea
Depressive symptoms Erectile dysfunction	Depressive symptoms Erectile dysfunction	
Low libido Tiredness	Difficulty concentrating Insomnia	
Loss of muscle mass Sexual dysfunction	Fatigue Increased body fat	
Difficulty concentrating Low libido	Depressed mood Irritability	
Erectile dysfunction	Lack of energy and sadness	
Low libido Tiredness	Fatigue Increased body fat	
Reduced energy Sexual dysfunction	Reduced energy and stamina Erectile dysfunction	
Sexual dysfunction	Memory impairment Headache	
Low libido Loss of muscle mass	Anxiety Irritability Insomnia	

## Data Availability

Not applicable.

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
