# Peer review of "Plasma Metabonomics in Insulin-Resistant Hypogonadic Patients Induced by Testosterone Treatment"

_ijms, 2022, doi:10.3390/ijms23147754_

Round 1
Reviewer 1 Report
In the manuscript “Plasma metabolomic profile in insulin-resistance hypogonadic patients induced by testosterone treatment”, the authors proposed to study the plasma metabolomics profile of insulin-resistance hypogonadic patients treated (or not) with testosterone. The topic of the manuscript is interesting and it could be of interest but there are too many drawbacks that hampered the enthusiasm. All sections need revision.
Specific comments:
1. Abstract needs to be rewritten. There is not information on the number of patients nor the data is well presented. Please provide a complete and relevant abstract.
2. To be accurate, the authors did a metabonomics study and not a metabolomics. This can be argued but by definition this is a metabonomic and not a metabolomics study.
3. The controls and the Hypogonadic patients have a very distinct BMI (25 vs 30). This is critical for data interpretation and must be well addressed in the paper since it can interfere with the results. Are these data really comparative? In addition, data from the patients should be the first results to be shown so that the reader can take conclusions regarding the quality of the paper in terms of clinical relevance. Table 1 should be moved.
4. Figure 1 is of very low quality and difficult to read. Please provide high quality images. It is hard to interpret these figures with such a low quality.
5. Line 132 – there is something missing here.
6. Figure 2 is also of low quality and the axis are difficult to read. There are some minor but important biochemical errors that must be corrected. For instance the use of NAD instead of NAD+.
7. It is very interesting that T treatment stimulates lactate content. It has been shown in rodent models that T deficiency induced by progressive stages of diabetes mellitus impairs glucose metabolism (PMID: 26148570). This could be briefly discussed by the authors.
8. There are too many typo and grammar errors that must be corrected. For instance, lines 157-158, the authors use 2 times, in 2 consecutive sentences “Interestingly”. Please carefully check the paper.
9. Figure 6 needs formatting (actually all figures, more or less need some work)
10. Discussion is very confusing and not really focused on the rationale for the experimental approach nor the major advancement attained by the data. The authors must present a more straightforward discussion of the hypothesis, with limitations well addressed and then a clear take home message. The clinical significance of the data must be also discussed.
Author Response
In the manuscript “Plasma metabolomic profile in insulin-resistance hypogonadic patients induced by testosterone treatment”, the authors proposed to study the plasma metabolomics profile of insulin-resistance hypogonadic patients treated (or not) with testosterone. The topic of the manuscript is interesting and it could be of interest but there are too many drawbacks that hampered the enthusiasm. All sections need revision.
Specific comments:
- Abstract needs to be rewritten. There is not information on the number of patients nor the data is well presented. Please provide a complete and relevant abstract.
We have modified the abstract according to the reviewer’s suggestions.
- To be accurate, the authors did a metabonomics study and not a metabolomics. This can be argued but by definition this is a metabonomic and not a metabolomics study.
We properly used the term “metabonomic” as suggested by the reviewer.
- The controls and the Hypogonadic patients have a very distinct BMI (25 vs 30). This is critical for data interpretation and must be well addressed in the paper since it can interfere with the results. Are these data really comparative? In addition, data from the patients should be the first results to be shown so that the reader can take conclusions regarding the quality of the paper in terms of clinical relevance. Table 1 should be moved.
As clearly reported, now, in the manuscript “although the BMI of hypogonadal subjects resulted to be higher than controls, the statistical analysis did not show significant differences in the three groups”
- Figure 1 is of very low quality and difficult to read. Please provide high quality images. It is hard to interpret these figures with such a low quality.
We modified the figure; figures have been moreover submitted as supplementary files in high-resolution format.
- Line 132 – there is something missing here.
We properly modifed the manuscript.
- Figure 2 is also of low quality and the axis are difficult to read. There are some minor but important biochemical errors that must be corrected. For instance the use of NAD instead of NAD+.
We modified the figure and corrected the errors; figures have been moreover submitted as supplementary files in high-resolution format.
- It is very interesting that T treatment stimulates lactate content. It has been shown in rodent models that T deficiency induced by progressive stages of diabetes mellitus impairs glucose metabolism (PMID: 26148570). This could be briefly discussed by the authors.
We discussed this topic according to your suggestion.
- There are too many typo and grammar errors that must be corrected. For instance, lines 157-158, the authors use 2 times, in 2 consecutive sentences “Interestingly”. Please carefully check the paper.
We modified the manuscript according to the reviewer’s suggestions. Two native English speakers revised and approved the final text.
- Figure 6 needs formatting (actually all figures, more or less need some work)
We modifed the figure; figures have been moreover submitted as supplementary files in high-resolution format.
- Discussion is very confusing and not really focused on the rationale for the experimental approach nor the major advancement attained by the data. The authors must present a more straightforward discussion of the hypothesis, with limitations well addressed and then a clear take home message. The clinical significance of the data must be also discussed.
Following the reviewer’s suggestion, we have modified the discussion and briefly reported the clinical significance of our results.
Reviewer 2 Report
Zolla et al., reported interesting data, especially for clinicians on metabolomic profiles in insulin-resistance patients with hypogonadism induced by testosterone treatment. However, the number of patients is low, preliminary new data are presented with the use of sophisticated analyses
1. Give detailed information about “testosterone plays a fundamental role in sexual, cognitive and body development…”.
2. “… increased libido, increased frequency of erections, increased muscle mass, increased voice, increased height, bone maturation “ ---increased or proper/normal/physiological?
3. IR-β, IRS-1, AKT-2, and GLUT4-provide full gene names
4. “Testosterone was introduced into the clinical setting for substitution therapy and its efficacy and safety in the treatment of male hypogonadism was demonstrated----give more details on therapy
5. “This has led to more testosterone synthesis options for physicians compared to injections”.-give details
6. Leyding cells??? “where one stimulates the production of the other and vice versa”—what production?
7. Testosteron, Lactate etc- capslock letter is not needed
8. Please write it like a discussion, not just information “The increased Acetyl-CoA was not even converted into mevalonic acid, which remained
low, and a minor cholesterol production was recorded (Table 1), in agreement with Zgliczynski et al. [41], as well as Acetyl-CoA was not used to produce acetylcarnitine, which decreased furtherly, blocking b-ossidation (oxidation) of FFA (Fig. 3 Panel A), in agreement with Fukami[42].
9. Please rewrite “ and 20 age and BMI-matched controls”----give information on age all of the patients together with important clinical data of hypogonadic ones
Author Response
- Give detailed information about “testosterone plays a fundamental role in sexual, cognitive and body development…”.
We better explained this sentence in the introduction section
- “… increased libido, increased frequency of erections, increased muscle mass, increased voice, increased height, bone maturation “ ---increased or proper/normal/physiological?
We modified this sentence according to the reviewer’s suggestions.
- IR-β, IRS-1, AKT-2, and GLUT4-provide full gene names
We modified the manuscript according to the reviewer’s suggestions.
- “Testosterone was introduced into the clinical setting for substitution therapy and its efficacy and safety in the treatment of male hypogonadism was demonstrated----give more details on therapy
We better discussed about testosterone therapy
- “This has led to more testosterone synthesis options for physicians compared to injections”.-give details
We modified the manuscript according to the reviewer’s suggestions and added informations about the options for testostosterone treatment.
- Leyding cells??? “where one stimulates the production of the other and vice versa”—what production? Testosteron, Lactate etc- capslock letter is not needed
We modified the manuscript according to the reviewer’s suggestions.
- Please write it like a discussion, not just information “The increased Acetyl-CoA was not even converted into mevalonic acid, which remained low, and a minor cholesterol production was recorded (Table 1), in agreement with Zgliczynski et al. [41], as well as Acetyl-CoA was not used to produce acetylcarnitine, which decreased furtherly, blocking b-ossidation (oxidation) of FFA (Fig. 3 Panel A), in agreement with Fukami[42].
We modified the discussion according to this suggestion.
- Please rewrite “ and 20 age and BMI-matched controls”----give information on age all of the patients together with important clinical data of hypogonadic ones
Changes have been done according the the suggestion of the Reviewer.
Round 2
Reviewer 1 Report
The paper can be accepted
Author Response
Thank you
Reviewer 2 Report
After succesful corrections in first review please add another ones:
1. "Testosterone was introduced into clinical settings as substitution therapy and its..."-give more details even these still controversial
2. Provide clear aim of the study
3. GLUT4-provide - full name
4. AKT serine/threonine kinase 2-check long name
Author Response
The changes were made according to the suggestions of the reviewers